# Is the Answer in the Text? Challenging ChatGPT with Evidence Retrieval from Instructive Text

**Sophie Henning**[1,2†]   **Talita Anthonio**[1,3†]   **Wei Zhou**[1,4]
**Heike Adel**[5]   **Mohsen Mesgar**[1]   **Annemarie Friedrich**[4]
[1]Bosch Center for Artificial Intelligence, Renningen, Germany
[2]Ludwig Maximilian University of Munich, Germany
[3]University of Stuttgart, Germany [4]University of Augsburg, Germany
[5]Hochschule der Medien Stuttgart, Germany
sophie.henning@de.bosch.com   talita.anthonio@yahoo.com
annemarie.friedrich@informatik.uni-augsburg.de
[†]Equal contribution

## Abstract

Generative language models have recently shown remarkable success in generating answers to questions in a given textual context. However, these answers may suffer from hallucination, wrongly cite evidence, and spread misleading information. In this work, we address this problem by employing ChatGPT, a state-of-the-art generative model, as a machine-reading system. We ask it to retrieve answers to lexically varied and open-ended questions from trustworthy instructive texts.

We introduce WHERE (**Wiki**How **E**vidence **RE**trieval), a new high-quality evaluation benchmark of a set of WikiHow articles exhaustively annotated with evidence sentences to questions that comes with a special challenge: All questions are about the article's topic, but not all can be answered using the provided context. We interestingly find that when using a regular question-answering prompt, ChatGPT neglects to detect the unanswerable cases. When provided with a few examples, it learns to better judge whether a text provides answer evidence. Alongside this important finding, our dataset defines a new benchmark for evidence retrieval in question answering, which we argue is one of the necessary next steps for making large language models more trustworthy.

## 1 Introduction

Generative language models (LMs) are trained to generate an output text given an input text. While such models have recently shown a remarkable performance in various NLP tasks (Touvron et al., 2023a; Radford et al., 2019; Brown et al., 2020), they are known to suffer from hallucination, i.e., they often generate text that lacks evidence (McKenna et al., 2023; Ji et al., 2022). This may lead to the spread of misinformation (Dong et al., 2022; Carlini et al., 2021), and thus reduce the systems' trustworthiness.

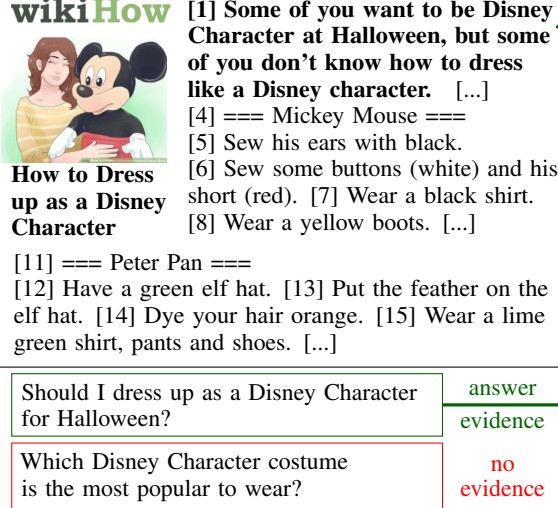

Figure 1: **Evidence retrieval for questions related to instructive articles from WikiHow**. For the question in the upper box, a system should ideally identify the sentences annotated as evidence in the text. For the question in the lower box, it should not retrieve wrong sentences as "evidence."

In this paper, we focus on a use case where a generative LM is queried for advice on a range of personal issues, including health or interpersonal relationships, or difficult tasks. This is a challenging scenario for LMs because questions are often open-ended and non-factoid, and require well-informed instructions as answers. As illustrated in Figure 1, we explicitly query generative LMs to retrieve evidence sentences for answering a question from a trustworthy instructive text. Our challenging setup requires two competencies on the model side: (i) identifying whether or not the question is answerable using only the provided text as input, and (ii) retrieving evidence from the trustworthy source, which could, e.g., support a generated answer.

Existing question answering datasets, e.g., Wiki-HowQA (Deng et al., 2020) and SQuAD (Rajpurkar

et al., 2016, 2018), do not fit our evaluation setup. The WikiHowQA dataset (Deng et al., 2020) uses the titles as questions, and does not cover the sentence retrieval aspect. SQuaD contains unanswerable questions but focuses on factoid questions. To make our evaluation setup challenging and sound, we create a new high-quality test set. We collect a set of diverse and open-ended questions for WikiHow articles via crowd-sourcing, and perform double annotation of evidence sentences in the articles. We use our dataset to perform a challenging evaluation of ChatGPT, a successor of InstructGPT (Ouyang et al., 2022b), which has been pretrained on a huge amount of texts including instructive web texts, in a systematic manner.

Our contributions are as follows: **(1)** We create and publish WHERE (**W**iki**H**ow **E**vidence **RE**trieval), a new high-quality evaluation test set of lexically diverse and open-ended questions for instructive articles taken from WikiHow. Evidence sentences are annotated with high agreement in all documents. WHERE contains both questions with evidence in the article and questions without. **(2)** We evaluate ChatGPT on this dataset in zero- and few-shot settings. Our experiments show that despite decent results on retrieving evidence for questions with evidence in the text, ChatGPT fails to recognize questions for which the text does not provide any evidence. When provided with a few no-evidence examples in the prompt, it refuses to answer if there is no evidence, but at the expense of recall of sentence retrieval. **(3)** We make our dataset and code publicly available.[1]

## 2 WikiHow Evidence Retrieval Dataset

Our goal is to collect questions with and without answer evidence in the text that are lexically, syntactically, and semantically diverse. In a pre-study, we find it hard to achieve this goal when the content of the article is known to the writer. We resort to crowdsourcing for question writing, and identify good cases by double-annotating the sentences deemed as evidence for the answer. Our reasoning is that if agreement on this task is low, the question is either somewhat ill-posed or too close to the overall topic of the article.

### 2.1 Dataset Creation

We export WikiHow[2] articles for the following categories:[3] *Arts and Entertainment*, *Home and Garden*, *Health and Relationships* and *Travel*. We collect articles for all categories in October 2022 and additional articles for *Home and Garden* and *Arts and Entertainment* in December 2022.

**Question collection.** To collect questions, we set up a Human Intelligence Task (HIT) on Amazon Mechanical Turk. We display the title of the article (e.g., "How to Dress up as a Disney Character"), the first paragraph of the article, and a set of keywords generated from the full article, and ask the crowd-workers to write six questions for which they would expect to find answers in the article (see Appendix D). Since annotators never see the full article, they can only make educated guesses about which questions may be answered by the full article and thus sometimes write questions that cannot be answered given the complete article. We encourage workers to start their questions with different question words (*what, why, how, can, should, who*). Our crowd-workers must be Master Workers, live in the UK or US, and have a HIT approval of at least 95%.

**Answer evidence annotation.** We tokenize documents into sentences using NLTK (Bird et al., 2009) and rule-based corrections, e.g., for enumerations. We then use the web-based annotation platform INCEpTION (Klie et al., 2018) to mark all sentences in an article that provide evidence for answering a question. We double-annotate 570 questions from 95 documents in two teams: one team is composed of several authors of this paper, the other team of paid annotators with engineering backgrounds and prior experience in NLP annotation tasks, who participate in a training phase. We allow annotators to discuss difficult cases within each team.

**Agreement.** We treat the annotations of one team as the gold standard and the other team's as the system. They agree with precision/recall/F1 of 70.6/57.3/63.3 on whether a document provides *no* evidence for a question at all. This corresponds to a $\kappa$-score (Cohen, 1960) of 0.43, which can be interpreted as *moderate* agreement according to Landis and Koch (1977).

To ensure a high-quality evaluation set, we com-

---

[1]https://github.com/boschresearch/where_emnlp_findings2023

[2]https://www.wikihow.com/Special:Export
[3]We thank WikiHow for granting us permission to redistribute the texts.

| Statistic | Value |
|---|---|
| # documents | 91 |
| # questions | 254 |
|    # with evidence in document | 129 |
|    # no evidence in document | 125 |
| # questions/document | $2.8 \pm 1.1$ |
| # evidence sentences/question* | $10.0 \pm 12.2$ |
| # sentences/document | $71.5 \pm 22.5$ |
| # tokens/document sentence | $16.4 \pm 8.9$ |
| # tokens/question | $13.6 \pm 3.5$ |

Table 1: Corpus statistics. *questions with evidence.

| Question Cat. | Count | % | Question Cat. | Count | % |
|---|---|---|---|---|---|
| how | 41 | 16.1 | which | 22 | 8.7 |
| what | 61 | 24.0 | who | 8 | 3.1 |
| when | 6 | 2.4 | why | 6 | 2.4 |
| where | 13 | 5.1 | yes/no | 97 | 38.2 |

Table 2: Distribution of question types in the test set.

pute question-level precision, recall, and F1 for the binary task of deciding whether a sentence provides evidence for answering a question. We keep only the questions with an F1 score of at least 0.3, and the questions that both teams consider to not have evidence. Annotators often disagree if the question is somewhat unclear or if the text contains only evidence of part of a question's aspects. By design, our filtering using a positive threshold for F1 removes any questions that only one team considered to have evidence in the article. Sentence-level $\kappa$ is 0.60, which is generally considered a solid score in semantic annotation tasks. For creating our gold standard, we take the union of the sentences marked by both teams as relevant evidence, as disagreements on the filtered cases are mostly due to different decisions on how much context to include.

## 2.2 Dataset Analysis

Table 1 shows the corpus statistics for our final test set. About half of the questions can be answered based on the evidence in the corresponding document (*with-evidence* questions), the other half cannot (*no-evidence* questions). Examples are shown in Figure 1. The instructive texts in WikiHow are kept simple as indicated by the short average sentence length. However, documents are long, which adds another challenge to our setup. For with-evidence questions, on average, about 14% of the sentences are part of the evidence.

Table 2 illustrates the distribution of question types: questions are indeed varied. Out of the yes/no questions, 81% ask for specific facts (*is/are there*, *will/do/does*) and 19% ask for suggestions or recommendations (*should/can/could*).

## 3 Method

We evaluate ChatGPT (version of March 2023, built upon GPT-3.5) in two settings. In the **zero-**

**shot setting**, we prompt ChatGPT using a template that asks the model to output a list of sentence IDs that can provide evidence for the answer. The list is empty if the model does not retrieve any evidence (see Appendix A). In the **few-shot setting**, we configure the message history such that ChatGPT sees five training examples from the test instance's category, e.g., *Health*, using the same prompt template as in the zero-shot setting but including gold responses. The training instances consist of additional question-article pairs annotated by the paid annotator team only. Every five-shot training set contains exactly two no-evidence questions and three with-evidence questions, which are about different documents.[4]

To accommodate for ChatGPT's context window size of 4096 tokens, we only use chunks of the training articles in the few-shot setting. Appendix B provides details.

Since ChatGPT may generate anything even though we ask it to output a list, we need to post-process the model outputs before evaluation. We first attempt to parse the model output using Python's eval function. If this fails, we try to match lists within the string (as ChatGPT sometimes provides explanations or just copies the selected sentences in addition to the list) using various regular expressions. If none of this works, which happens only three times in the entire experiment, we manually parse the model output. A small number of questions is rejected by OpenAI's content filter. For those, we assume the empty list as model output as ChatGPT does not retrieve any evidence from the article.

## 4 Experiments and Analysis

**Evaluation metrics.** For no-evidence questions, we report recall for correctly recognizing that an article does not contain any evidence for answering a question. For with-evidence questions, we compute precision, recall, and F1 score of the rele-

---

[4]We also attempted manual prompt engineering, leading to similar results.

vant class ("evidence sentence") per question, and report macro-average over questions.

**Baselines.** The random baseline for with-evidence questions predicts that a sentence contains relevant evidence in 14% of the cases, which corresponds to the average percentage of sentences marked as evidence per question in the test set. Similarly, it predicts "no-evidence" for a question with a probability of 49.2%. The results reported for "human scores" correspond to the agreement scores.

**Results.** Table 3 shows the zero- and few-shot performance of ChatGPT, separately evaluated on with-evidence and no-evidence questions. In the zero-shot setting, ChatGPT outperforms the baseline for the with-evidence questions by a large margin, but fails to recognize when there is no evidence for a question in an article. When presented with five training instances of which two are no-evidence questions, ChatGPT almost reaches human performance on recognizing no-evidence questions. This, however, comes at the cost of decreased performance on the questions with evidence. On with-evidence questions, ChatGPT is still far from human performance in both zero- and few-shot settings.

**Analysis.** In the zero-shot setting, OpenAI's content filter identifies the content of 10 of the test cases harmful and declines to generate outputs for them. We do not consider these instances to be harmful, e.g., they are about instructions on how to find emergency procedures in hotels. For the few-shot configurations, these cases vary from 11 to 16 instances, depending on the few-shot prompts. This illustrates that improving content filters is an important future research direction.

Table 4 breaks down the evaluation results by the question types identified in Table 2. Comparing *how-*, *what-* and yes/no questions, we find that the latter are the easiest in both the zero- and the few-shot setting. In terms of recognizing that there is no evidence for a question, *how* questions are consistently more difficult than *what* and yes/no questions.

**Applicability beyond ChatGPT.** While this paper focuses on ChatGPT, WHERE can be used to evaluate any other large LM whose training data does not contain WHERE. We also evaluated Llama 2-Chat (Touvron et al., 2023b) in the zero-shot setting, finding it to perform considerably worse than ChatGPT in the same setting (see

| Model | With-evidence questions | | | Recall |
| | mac.P | mac.R | mac.F1 | no-evidence |
|---|---|---|---|---|
| random baseline | 14.0 | 14.0 | 11.3 | 49.2 |
| ChatGPT 0-shot | 37.9 | **52.6** | **37.7** | 7.2 |
| ChatGPT 5-shot | **38.2** | 42.8 | 35.5 | **60.3** |
| | ± 2.0 | ± 2.1 | ± 0.9 | ± 5.5 |
| "human" scores | 64.4 | 77.0 | 64.2 | 63.3* |

Table 3: Results on our evaluation set. For 5-shot, we report average and standard deviation of 5 different randomly sampled per-category 5-shots results. *Harmonic mean of the two recall scores, estimated on all 570 double-annotated questions.

Table 5 in Appendix C). In addition, we needed to manually parse 30 model outputs (compared to one when using ChatGPT).

**Impact of pre-training data.** It is well possible that some of the WikiHow articles (or earlier versions thereof) have been part of ChatGPT's training data. For our evaluation setup, however, this is not an issue, since the documents are contained in the model input anyway. We intentionally did not crawl questions from the web or, e.g., the WikiHow comments sections, but let crowd-workers write them based on our methodology mitigating bias towards the articles content. Hence, it is unlikely that existing LLMs have seen these specific questions or their answers as part of their pre-training.

## 5 Related Work

**Extractive QA datasets.** The SQuAD datasets (Rajpurkar et al., 2016, 2018) are the largest datasets for extractive QA. The questions are factoid and can be answered through short, single answer spans. In contrast, our dataset includes non-factoid questions based on instructional text and requires models to identify a set of answer spans across long documents. The MultiSpanQA (Li et al., 2022) and MASH-QA (Zhu et al., 2020) datasets also contain QA pairs with several answer spans. However, MultiSpanQA is automatically derived from Natural Questions (Kwiatkowski et al., 2019) and MASH-QA also contains automatically created QA pairs. Only few datasets contain unanswerable questions, e.g., the latest version of SQuAD. However, its questions have been created given a text passage, whereas the questions of WHERE are based only on the keywords and a summary of an article. This leads to larger lexical variety between the question and the text and, thus, creates a more realistic and challenging setting.

| Type (#w/#n) | Setting | With-evidence questions | | | Recall no-evidence |
|---|---|---|---|---|---|
| | | mac.P | mac.R | mac.F1 | |
| how (21/20) | 0-shot | 38.1 | **55.0** | **35.1** | 0 |
| | 5-shot | **38.6** | 40.2 | 33.8 | **42.0** |
| | | ± 5.0 | ± 3.1 | ± 3.9 | ± 7.6 |
| what (41/20) | 0-shot | 31.4 | **48.7** | **32.1** | 10.0 |
| | 5-shot | **35.9** | 39.6 | 31.0 | **56.0** |
| | | ± 2.9 | ± 2.1 | ± 3.8 | ± 8.2 |
| when (0/6) | 0-shot | - | - | - | 0 |
| | 5-shot | - | - | - | **43.3** |
| | | | | | ± 14.9 |
| where (7/6) | 0-shot | **48.3** | 51.4 | **46.9** | 16.7 |
| | 5-shot | 35.2 | **52.3** | 35.9 | **60.0** |
| | | ± 8.2 | ± 20.7 | ± 11.9 | ± 9.1 |
| which (13/9) | 0-shot | **44.5** | **58.6** | **47.0** | 0 |
| | 5-shot | 41.1 | 47.7 | 39.7 | **68.9** |
| | | ± 4.4 | ± 3.8 | ± 2.8 | ± 12.2 |
| who (2/6) | 0-shot | 21.4 | **100** | 34.7 | 0 |
| | 5-shot | **67.7** | 95.0 | **75.2** | **80.0** |
| | | ± 5.0 | ± 11.2 | ± 6.3 | ± 7.5 |
| why (2/4) | 0-shot | 12.5 | **50** | 20.0 | 25.0 |
| | 5-shot | **20.0** | 40.0 | **26.7** | **30.0** |
| | | ± 11.2 | ± 22.4 | ± 14.9 | ± 11.2 |
| yes/no (43/54) | 0-shot | **42.4** | **51.5** | **41.0** | 9.3 |
| | 5-shot | 39.4 | 41.2 | 36.8 | **69.3** |
| | | ± 4.9 | ± 5.0 | ± 2.6 | ± 7.6 |

Table 4: Results of ChatGPT on our evaluation set, separated by question type. For 5-shot, we report average and standard deviation of 5 different randomly sampled per-category 5-shots results. -: no questions of this type in the dataset. #w: number of with-evidence questions, #n: number of no-evidence questions.

**Evidence retrieval.** Our work builds on a long line of NLP work addressing the fine-grained retrieval of supporting facts from text. This is a common basis for tasks like question answering (Murdock et al., 2012), fact checking, or slot filling (Petroni et al., 2021), and has also been used to support other tasks, e.g., question generation (Lewis et al., 2020), document retrieval (Akkalyoncu Yilmaz et al., 2019) or natural language inference (Vladika and Matthes, 2023).

**Analysis of ChatGPT.** Since the publication of ChatGPT, many papers focus on analyzing the applicability of the system (Liu et al., 2023b) for different tasks and domains, i.a., question answering for education (Frieder et al., 2023) and in medical environments (Nov et al., 2023). This large attention indicates the importance of the system for real-world applications and is the main motivation for our work. Despite promising performance in several benchmark datasets, for instance logical reasoning comprehension (Liu et al., 2023a)

or complex question answering (Tan et al., 2023), previous studies also reveal short-comings of GPT models, for instance, their hallucination problem (Ouyang et al., 2022a; Li et al., 2023; Zhang et al., 2023). To address this issue, we propose to analyze the usage of ChatGPT in a setting in which we force it to retrieve evidence from text instead of directly generating an answer.

## 6 Conclusion and Outlook

In this work, we have introduced a challenging benchmark to evaluate ChatGPT on the task of retrieving evidence sentences from instructive text to answer a question. We have presented a new high-quality dataset from WikiHow consisting of crowd-sourced questions and expert evidence annotations. A special real-world challenge of our dataset is the inclusion of questions that are not answerable given an article. When evaluating ChatGPT on our dataset, we found that it fails on detecting no-evidence questions if not provided with targeted examples in the prompt. Our results on the new benchmark highlight important shortcomings of generative large language models that need to be addressed by future research: besides hallucinating answers, there is no free lunch with regard to calibration in evidence-retrieval setups.

## Limitations

Since we accessed ChatGPT via the OpenAI API, our results might not be reproducible if the model behind the API is exchanged with a new version. Moreover, previous work has shown that ChatGPT is stable only to an extent of 79% for complex question answering tasks (Tan et al., 2023).

Another limitation of using ChatGPT via the API is that we do not have access to the model parameters and probability distributions. This reduces the amount of analysis we can perform on the model results.

The dataset we present in this work is double-annotated and, as a result, of high quality but rather small compared to other question-answering datasets (but other datasets are often crowd-sourced, created using heuristics, and/or contain less varied questions).

## Ethics Statement

**Data source and licensing.** We ensured that we may re-publish the WikiHow texts under CC BY-NC-SA-3.0 by obtaining written permission from

WikiHow. Some of the topics of the selected articles are about health and relationships, yet, there is no personal information involved.

**Crowd-sourcing and annotation.** We collect our set of questions via Amazon Mechanical Turk. We pay 1 dollar per HIT, but increase to 2 dollars/HIT in the subsequent tasks, given the average completing time (between 5 and 10min).

The paid annotators participating in our project were completely aware of the goal of the annotations and even helped designing the annotation scheme. They gave explicit consent to the publication of their annotations. The main annotator was paid considerably above our country's minimum wage. For this type of semantic annotation task not involving any personal data, our institution did not require obtaining an IRB Review.

## Acknowledgements

We thank Dragan Milchevski and Zhe Feng for helpful discussions on the few-shot setting.

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

## Supplementary Material

## A Prompt Template

Figure 2 shows the complete message histories passed through ChatGPT to generate system outputs.

Our prompt template for ChatGPT is as follows:

**System message**: `Your task is to select sentences from a document that answer a given question.`

**User message (question, document)**: `Select sentences from the document below that answer the question below. It may also be the case that none of the sentences answers the question. In the document, each sentence is marked with an ID. Output the IDs of the relevant sentences as a list, e.g., "[1,2,3]", and output "[]" if no sentence is relevant. Output only these lists.`
`Question:"'<question>"'`
`Document:"'<document>"'`

## B Sampling Chunks for Few-Shot Instances

To fit ChatGPT's maximum input size, we only provide chunks of the few-shot training instances. [5] We use the tiktoken library[6] to count the number of tokens needed for prompting ChatGPT and create chunks accordingly. The largest test input instance requires 2235 tokens, i.e., 1861 tokens remain for the few-shot instances and the model output, called "completion" by OpenAI. In the ground-truth answers, the maximum number of completion tokens is 189. To enable some flexibility of the model, we reserve 300 tokens for completion, which equals the amount of tokens needed to encode a list of sentence IDs from 0 to 99 including. Hence, 1561 tokens remain to encode the 5 training instances, i.e., approximately 312 tokens per training instance (including the ground-truth answer). We fill these tokens from the training instances by sampling a context window of at most 350 tokens around a random (relevant in case of a question with evidence) sentence.

---

[5]To fit entire documents, we would have needed to use the 32k version of GPT-4, which is up to 60x more expensive than ChatGPT.

[6]https://github.com/openai/tiktoken

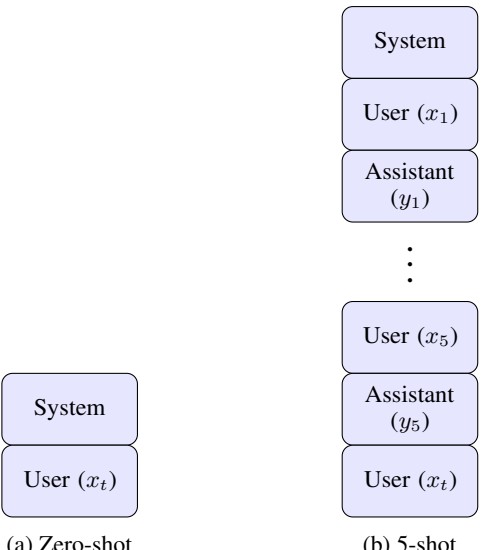

(a) Zero-shot     (b) 5-shot

Figure 2: **Message history** passed as input to ChatGPT in zero- and few-shot setting. $x_t$: test instance, $x_i$: training instance $i$, $y_i$: ground truth for $x_i$.

| Model | With-evidence questions | | | Recall no-evidence |
|---|---|---|---|---|
| | mac.P | mac.R | mac.F1 | |
| random baseline | 14.0 | 14.0 | 11.3 | 49.2 |
| ChatGPT 0-shot | 37.9 | **52.6** | **37.7** | 7.2 |
| Llama 2 0-shot | 21.9 | 23.7 | 18.6 | 6.4 |
| ChatGPT 5-shot | **38.2** | 42.8 | 35.5 | **60.3** |
| | ± 2.0 | ± 2.1 | ± 0.9 | ± 5.5 |
| "human" scores | 64.4 | 77.0 | 64.2 | 63.3* |

Table 5: Results on our evaluation set. For 5-shot, we report average and standard deviation of 5 different randomly sampled per-category 5-shots results. *Harmonic mean of the two recall scores, estimated on all 570 double-annotated questions. Llama 2: Llama 2-Chat (13B).

## C Additional Results

Here, we provide an extended version of Table 3 that includes scores for Llama 2-Chat (13B) (Touvron et al., 2023b) in the zero-shot setting. Compared to ChatGPT, LLama 2 is equally bad at detecting no-evidence questions and recognizes relevant sentences for answering with-evidence questions worse.

## D Crowdsourcing

We show the interface that we used for collecting the questions on Amazon Mechanical Turk in Figure 3. The interface contains two parts. On the left, we show crowd-workers two components based on the WikiHow article: the summary and the keywords. The summary is the first paragraph from a

WikiHow article, which typically functions as an introduction to the article's topic. The keywords contain a set of keywords that we generated based on the WikiHow article using the `Wordcloud`[7] package. On the right, we provide a form where crowd-workers can submit six questions. Crowd-workers are encouraged to make use of our suggestions on how to start a question and to start each question with a different word. Furthermore, in the first three input fields, we ask crowd-workers to ask a question about a certain topic. In Figure 3, these topics are, from top to bottom: *Researching Upgrade Options*, *Planning Your Honeymoon* and *Upgrading on the go*. These are the section headers from the WikiHow article, which we use to collect questions that are relevant to the article.

---

[7] https://pypi.org/project/wordcloud

## Summary

**How to Get Free Honeymoon Upgrades**

Everyone loves free upgrades during trips, and a honeymoon is a perfect excuse to enjoy them.

With a little research, some strategic planning, and a willingness to share your newlywed status on-the-go, you're sure to collect some luxurious upgrades.

Just maintain a kind, enthusiastic attitude and get ready for an upgraded honeymoon of a lifetime!

## Keywords

The keywords (verbs and other words) below are related to **How to Get Free Honeymoon Upgrades**. Use them in your questions.

**Verbs:**

make, ask, offer, give, may, mention, try, sign, take, upgrade, will, call, bring, love, help, include, look, book, want, arrive, dress, enjoy, share, add, gift, start, plan, provide, qualify, search, talk, know, go, request, reserve, say, show, feel, tip, collect, maintain, lead, allow, encourage, spend, use, register, complete, check, consider

**Other words:**

honeymoon, upgrade, free, car, staff, card, company, travel, perk, hotel, special, small, incentive, sure, mile, flight, good, reservation, compact, frequent, flier, credit, rental, way, likely, airport, book, chance, restaurant, seat, gift, trip, airline, program, lot, friendly, wedding, well, coupon, destination, demand, season, agent, person, people, extra, important, favor, tip, everyone

## Questions

**Suggestions on how to start questions:**

- How can/should I , When, Why, What, Where, Which, Who
- Do I ... , Can I , Is it ... , Should I

**Question 1**

Ask question about Researching Upgrade Options

**Question 2**

Ask question about Planning Your Honeymoon. Use different start word.

**Question 3**

Ask a question about Upgrading on the Go. Use different start word.

**Question 4**

Ask a question. Use different start word.

**Question 5**

Ask a question. Use different start word.

**Question 6**

Ask a question. Use different start word.

Figure 3: The interface used in our crowdsourcing set-up for collecting questions. The left part of the interface shows the summary and keywords based on the WikiHow article and the right shows the form that crowd-workers used to submit their questions.

## E  Dataset Example

**Title: How to Find Trusted Advice on Covid-19**

**Questions:**

1. Is there a central authority that I should check with for COVID-19 advice?
   [5, 6, 7]

2. What groups does the most research on COVID-19?
   *no-evidence*

3. How can I know if I'm hearing the truth about COVID-19 when listening to someone?
   [18, 23, 27, 28, 32, 38, 39]

4. Who should I follow to find out about the latest COVID-19 variants?
   *no-evidence*

**Article Text:**

[1] Whether you're surfing the web, texting a friend, or tuning into the nightly news, you're probably hearing a lot of different things about the COVID-19 outbreak.

[2] It's difficult to get a finger on the pulse of what's going on during the current state of the world, but there are several ways to make the fact-checking process a bit easier.

[3] If you know where to look, and subsequently, where to stay away from, you can stay informed as the COVID-19 situation continues to develop.

[4] ==Steps==

**[5] === Reliable Organizations===**

**[6] Consult WHO and the UN for reliable, global updates.**

**[7] The World Health Organization (WHO) and the United Nations (UN) are constantly studying and reporting on COVID-19 cases all over the world.**

[8] These organizations' websites offer plenty of resources and articles that you can peruse, which can help keep you up-to-date on the latest news and best practices for staying safe during the pandemic.

[9] * You can find a list of common COVID-19 mythbusters here:

[10] Visit the COVID-19 "hubs" on various social media and internet platforms.

[11] Sites like Facebook, Apple News, Google, Snapchat, and Twitter have all created special "hubs," or featured sections of information pertaining to the COVID-19 outbreak.

[12] You'll have to use a bit of your own discretion as you go through the different news bytes—however, many of these platforms try to prioritize more reliable news sites.

[13] * These are the easiest ways to stay up-to-date with the newest COVID-19 developments.

[14] Stop by the Johns Hopkins site for accurate reports of case numbers.

[15] It can be a bit gloomy to think about how many COVID cases there are in the world currently.

[16] However, if you want a more exact count, the Johns Hopkins Center for Systems Science and Engineering runs a dashboard that calculates the current number of global cases.

[17] * You can find this site here:

**[18] Follow trusted experts on social media for factual information.**

[19] Platforms like Twitter can be rife with misinformation if you're following the wrong people, but they can be a great source of news and factual information if you're following medical experts.

[20] * Doctors and members of the medical community are great people to listen to during the pandemic.

[21] ===Unreliable Information Sources===

[22] Don't put too much stock in random social media posts.

**[23] The current climate has left a lot of people nervous and anxious about the coming days, which is perfectly understandable.**

[24] It can be easy to believe what you see on social media, but take the time to scrutinize posts carefully and discredit any info you find about home-brewed cures or solutions for COVID-19.

[25] Search for studies only published by reliable groups.

[26] If you're browsing through different studies, look over the specific publication details.
**[27] While many studies are authoritative, there can be different factors that impact a study's credibility, like the funding source, or where the study was first published.**
**[28] Generally, try to get your information from studies published in reputable journals.**
[29] Fact-check new information against reputable organizations.
[30] Studies are teaching us new things every day, but they aren't rewriting the rules when it comes to COVID-19, either.
[31] Even if new information is released in a study, you shouldn't toss out everything you've learned out the window, either.
**[32] Try to fact-check new information against reputable organizations, so you can get an idea of what to believe.**
[33] * For instance, you can cross-check new studies with WHO.
[34] Report misinformation as you find it online.
[35] It can be really frustrating to find blatantly incorrect things online.
[36] Thankfully, most sites and platforms give you the option to report misinformation, which can help make the online world a safer place.
[37] You can figure out the best way to report misinformation here:
**[38] == Tips ==**
**[39] *The best way to stay informed is to cross-check new information over several reliable sources.**
[40] *If a source follows a strict fact-checking process, it's probably a safe source of information.
[41] ==Warnings==
[42] *Don't fall for the narrative that certain demographics are more "likely" to believe false facts or spread misinformation.

**Discussion:**

An example for a clear case is provided for question 1: sentence [6] "Consult the WHO and UN for reliable, global updates." clearly provides evidence for the answer to question 1.

Answering question 3 based on the text is possible, but requires more reasoning. For example, sentence [18] can be considered as evidence in the sense of "if an expert is trusted, you will get factual information from them, and know that you're hearing the truth about COVID-19." Sentence [23] is a borderline case (which this author annotator would not have marked), but we see the interpretation that one should rather distrust nervous and anxious people.

Both annotator teams agree that the article does not provide any evidence for questions 2 and 4.