# OpenReview forum: "Is the Answer in the Text? Challenging ChatGPT with Evidence Retrieval from Instructive Text"
_EMNLP/2023/Conference — EMNLP 2023 Findings_

### Official Review · Reviewer_zJHM · 2023-08-04

**Soundness:** 3

**Excitement:**

3: Ambivalent: It has merits (e.g., it reports state-of-the-art results, the idea is nice), but there are key weaknesses (e.g., it describes incremental work), and it can significantly benefit from another round of revision. However, I won't object to accepting it if my co-reviewers champion it.

**Paper Topic And Main Contributions:**

This paper proposes a new dataset for question-answering, which contains unanswerable questions. Unlike previous similar datasets, the proposed dataset also includes non-factoid questions and requires the system to identify evidence sentences in a given text or return no answer (i.e., no-evidence). The paper uses WikiHow articles as input texts and asks human annotators to generate various question types (e.g., how, what, when, etc.). The evaluation is focused on ChatGPT with two settings: zero-shot and few-shot. The paper finds that ChatGPT fails to recognize no-evidence questions in the zero-shot setting. This issue can be alleviated in the few-shot setting but at the cost of recall of with-evidence questions.

Strengths
- The problem setting is interesting (i.e., unanswerable and non-factoid questions).
- The dataset was well-designed and carefully constructed.

Weaknesses
- While the dataset would be used in subsequent work, I am unsure whether the findings about ChatGPT's behaviors are useful for the research community.
- The random baseline is weak.

Comments
- In the analysis (lines 261-265), it would be better if the paper also shows the statistics per question type.
- The case when ChatGPT fails to return no evidence can be expected because it (as an LLM in general) is pre-trained to predict the next token given the context input.
- The paper would consider adding the actual ChatGPT's outputs in the appendix.

**Questions For The Authors:**

Would the results be affected if WikiHow articles were included in ChatGPT's training data?

**Reasons To Accept:**

The proposed dataset can complement existing ones with a wide range of unanswerable/non-factoid questions.

**Reasons To Reject:**

The findings are limited to one generative language model, ChatGPT, and might only be valid with the ChatGPT version the paper experimented with.

**Reproducibility:**

3: Could reproduce the results with some difficulty. The settings of parameters are underspecified or subjectively determined; the training/evaluation data are not widely available.

**Reviewer Confidence:**

3: Pretty sure, but there's a chance I missed something. Although I have a good feel for this area in general, I did not carefully check the paper's details, e.g., the math, experimental design, or novelty.

---

> ### Author Rebuttal · Authors · 2023-08-28
>
> Thank you for reviewing our paper, and for calling our dataset “well-designed and carefully constructed” and recognizing its complementariness to existing datasets.
>
> Comments on weaknesses / reasons to reject:
>
> - Concern on the usefulness of our findings on ChatGPT for the research community: ChatGPT (GPT-3.5)’s outputs are often used as a teacher to train open-source model (see, e.g., https://arxiv.org/pdf/2304.12244.pdf, https://lmsys.org/blog/2023-03-30-vicuna/). Thus, analyzing the output of models like ChatGPT is necessary for the research community, with our finding on ChatGPT’s behavior with respect to no-evidence question contributing an important aspect to the bigger picture.
> - For a discussion on the random baseline, please see our response to Vngq: Our point is that **the random baseline actually outperforms ChatGPT** on the task of recognizing whether or not a text provides evidence for answering a question.
>
> Other comments:
> - Thank you for the great suggestion to add evaluation results per question type. We will add them in the camera-ready version.
> - We agree that it can be expected to some degree that ChatGPT fails to return no evidence. However, the prompt we use in the experiments explicitly tells it to output “[]” if there is no evidence in the text. From a human and an application perspective, noticing that one does not know something is crucial, and should also be required from LLMs. We have established that ChatGPT fails this requirement with respect to missing evidence in instructional texts.
> - Thank you for your suggestion to add ChatGPT’s outputs in the appendix. We will make the outputs publicly available.
>
> Question on effects of WikiHow articles potentially being included in ChatGPT’s training data: As the articles are anyways included in the prompt, we believe that whether or not some of the articles have been part of ChatGPT’s training data does not affect our results.

---

### Official Review · Reviewer_cu6q · 2023-08-04

**Soundness:** 3

**Excitement:**

3: Ambivalent: It has merits (e.g., it reports state-of-the-art results, the idea is nice), but there are key weaknesses (e.g., it describes incremental work), and it can significantly benefit from another round of revision. However, I won't object to accepting it if my co-reviewers champion it.

**Paper Topic And Main Contributions:**

The paper primarily focuses on evaluating ChatGPT's performance in the context of evidence retrieval from WikiHow articles. The main contributions of the paper are:

1. Creation and publication of a new high-quality evaluation test set comprising lexically diverse and open-ended questions for instructive articles.
2. sourced from WikiHow. This dataset includes both questions with evidence in the article and questions without.
3. A systematic evaluation of ChatGPT on this dataset in both zero- and few-shot settings. The results highlight ChatGPT's strengths and weaknesses in recognizing evidence-based and no-evidence questions.

**Questions For The Authors:**

How do you see the application of this evaluation method to other large language models beyond ChatGPT?

**Reasons To Accept:**

1. The paper addresses a pertinent issue in the NLP community, which is the evaluation of large language models like ChatGPT in real-world scenarios.
2. The creation of a new dataset specifically tailored for this task is commendable. The dataset's focus on lexically diverse and open-ended questions makes it a valuable resource for future research.
3. The systematic evaluation provides insights into the capabilities and limitations of ChatGPT, which can guide future improvements in the model.

**Reasons To Reject:**

The dataset, while of high quality, is relatively small compared to other question-answering datasets.

**Reproducibility:**

3: Could reproduce the results with some difficulty. The settings of parameters are underspecified or subjectively determined; the training/evaluation data are not widely available.

**Reviewer Confidence:**

3: Pretty sure, but there's a chance I missed something. Although I have a good feel for this area in general, I did not carefully check the paper's details, e.g., the math, experimental design, or novelty.

---

> ### Author Rebuttal · Authors · 2023-08-28
>
> Thank you for reviewing our paper, and for appreciating our new resource and the insights on “capabilities and limitations of ChatGPT” that our evaluation provides.
>
> We agree with you that our dataset is relatively small compared to other QA datasets, but it is of high quality (please also see the discussion in the Limitations section). The dataset is large enough for the focused contributions of this short paper: demonstrating the capabilities and weaknesses of ChatGPT-like models when retrieving evidence from instructive texts.
>
> Regarding your question on the application of our evaluation method to other LLMs beyond ChatGPT: Any LLM can be tested with our dataset (given that it is not comprised in the LLM’s training data). The methodology that we propose to collect high-quality testing data can be used to create further data, also in other genres/domains. Constructing high-quality test sets has become an important goal, as the training data of LLMs is often not disclosed.

---

### Official Review · Reviewer_Vngq · 2023-08-05

**Soundness:** 3

**Excitement:**

2: Mediocre: This paper makes marginal contributions (vs non-contemporaneous work), so I would rather not see it in the conference.

**Missing References:**

The paper focuses on evidence retrieval. However, there's no discussion of related works on this task.

**Paper Topic And Main Contributions:**

This paper investigates ChatGPT's ability for evidence retrieval. The contribution includes developing a new evaluation dataset that comprises questions with evidences and questions without. In addition, the author prompts ChatGPT to retrieve evidences from a given document to the question. The model also needs to recognize when there's no evidence for answering the questions. Experimental results including a zero-shot and 5-shot setting demonstrate that ChatGPT outperform a random baseline.

**Questions For The Authors:**

1. Which backbone of ChatGPT is evaluated?
2. How do you split the document into numbered sentences, as shown in Figure 1?
3. l. 116, if the annotators are asked to write questions that can be answered by the document, then where are the no-evidence questions coming from?
5. l. 253 - l. 258, are there any examples to illustrate the failed cases?

**Reasons To Accept:**

1. The idea of using ChatGPT (or other LLMs) for evidence retrieval is sound.
2. The evaluation of ChatGPT provides discoveries and insights that may benefit future works.
3. The paper present new resources with a new dataset.

**Reasons To Reject:**

1. The conclusion is not well-supported. There's no persuasive baselines since the only one being compared is a random baseline.
2. The work is fairly simple, the author provides with limited analysis of the results and no extra discussions or ablation studies.
3. The final dataset comprises only 254 questions from 91 documents and includes only a few samples for 3 question types, I am not convinced that the dataset is high-quality and lexically diverse.
4. The paper needs careful proofreading and revising, as many details are missing or explained unclearly. Please refer to the question section.

**Reproducibility:**

4: Could mostly reproduce the results, but there may be some variation because of sample variance or minor variations in their interpretation of the protocol or method.

**Reviewer Confidence:**

4: Quite sure. I tried to check the important points carefully. It's unlikely, though conceivable, that I missed something that should affect my ratings.

**Typos Grammar Style And Presentation Improvements:**

Typos:
l.181, "of" → "if"

---

> ### Author Rebuttal · Authors · 2023-08-28
>
> Thank you for reviewing our paper, and for valuing our new resource. As the other reviewers mention, in addition to contributing a new dataset and methodology to construct such datasets, the important main finding of our paper is that ChatGPT almost completely fails to recognize the no-evidence questions in the zero-shot setting.
>
> Comments on the reasons to reject you list:
>
> 1.	Thanks for suggesting to report further baselines (e.g., a fine-tuning baseline) to the experiments. We also ran a preliminary check on Llama 2 (13B), finding it to perform considerably worse than ChatGPT, and are happy to add these scores to the paper. However, please note that our point is not that ChatGPT outperforms a random baseline, but that the **random baseline actually outperforms ChatGPT when it comes to recognizing questions that cannot be answered based on a text** (last column of Table 3)! Contrasting the 0-shot and the 5-shot rows, we see that ChatGPT can be taught to realize this, but at the cost of correctly answering the with-evidence questions.
>
> 2.	We agree that the experiments in our paper are relatively straightforward. However, the development of the methodology was not, and necessitated a number of experiments and annotation steps. We ensured question diversity in the first step of the dataset collection using a carefully designed crowdsourcing interface (see Figure 3 in the Appendix) and then doubly-annotated the dataset in a second step to filter the dataset for high-agreement cases. We argue that our **short** paper contains a sufficient amount of discussion and analysis of the issue of realizing whether a text does or does not contain evidence for a question in the context of real-world problems and LLMs.
>
> 3.	On the size of the dataset, please see the discussion in our Limitations section. It is true that for three question types (when, who and why) there are fewer instances, but for the other diverse question types (how, where, which, yes/no), we collected a larger number of questions. We believe that the dataset itself and the associated methodology provide enough grounding for future research. We will follow your suggestion to add more examples to the paper in order to illustrate the lexical diversity of our dataset. Please note that the entire dataset will also be released. Our sentence-level agreement (Cohen’s kappa) is 0.6, which is considered as moderate (almost substantial) agreement according to Landis and Koch (1977). In the context of semantic annotation tasks, this is usually considered as sufficient agreement.
>
> 4.	Thank you for this feedback, the questions you provide are easy to clarify and we will gladly add the requested information to the paper.
>
> Answers to your questions:
>
> 1.	We use ChatGPT with GPT-3.5 (version of March 2023) as its backbone.
>
> 2.	The sentence tokenization is based on NLTK and rule-based corrections (e.g., for enumerations).
>
> 3.	For each article, annotators only see the title and the first paragraph (see Section 2.1). Additionally, they get a set of keywords generated from the full article, but they never see the full article. Hence, they can only make educated guesses about which questions may be answered by the full article (cf. l.111-l.117 and Figure 3) and thus sometimes write questions that cannot be answered given the complete article.
>
> 4.	We will use the extra page in the camera-ready version to include these examples in the paragraph starting at line 252 and potentially add more to the appendix.
>
> Missing References:
>
> Thank you for this suggestion. As our short paper focuses on the machine reading component, we mainly discussed this in the related works section. We will gladly add some references on evidence retrieval

---

### Meta-Review · Area_Chair_XRLa · 2023-09-17

**Recommendation:** 3

**Metareview:**

The reviewers agree this work proposes a useful evaluation resource, and a surprising negative result in how poorly ChatGPT performs on “no-evidence” questions.
While the analysis is limited and the findings are only for ChatGPT, for a short paper this work provides useful resources and informative insights to practitioners using ChatGPT for question answering in ambiguous settings.

---

### Decision · Program_Chairs · 2023-10-07

**Decision:**

Accept-Findings

**Comment:**

The reviewers agree this work proposes a useful evaluation resource, and a surprising negative result in how poorly ChatGPT performs on “no-evidence” questions.
While the analysis is limited and the findings are only for ChatGPT, for a short paper this work provides useful resources and informative insights to practitioners using ChatGPT for question answering in ambiguous settings.